# Intra-city Scale Graph Neural Networks Enhance Short-term Air Temperature Forecasting

Han Wang[1], Jianheng Tang[2], Jize Zhang[1], Jiachuan Yang[1]

[1]Department of Civil and Environmental Engineering, The Hong Kong University of Science and Technology, Hong Kong, 999077, China

[2]Division of Emerging Interdisciplinary Areas, The Hong Kong University of Science and Technology, Hong Kong, 999077, China

*Correspondence to*: Jiachuan Yang (cejcyang@ust.hk)

**Abstract.** Air temperature ($T_a$) has critical implications for various socioeconomic sectors, yet its dynamics are particularly complex in urban areas due to heterogeneous built environments, landscapes, and diverse anthropogenic activities. Physics-based models struggle with intra-city $T_a$ forecasts due to inadequate urban representation and limited spatial resolution. While weather observation networks offer promising alternatives for direct modeling with local $T_a$ time-series, an effective framework to leverage these intra-city discrete sensor data remains lacking. Here, we demonstrate that graph neural networks (GNNs) can harness observation network information to refine $T_a$ prediction at individual locations and elucidate underlying mechanisms. Our novel Mix-n-Scale framework with GNNs achieves over 12% improvement in short-term $T_a$ forecasts compared to conventional local time-series approaches. Further model evaluation disentangles performance variations with local $T_a$ variability in diverse spatiotemporal contexts, indicating distinct patterns of intra-city heterogeneity across seasonal and diurnal scales. Our findings establish graph-based approaches for leveraging proliferating urban sensor data and advancing understanding of $T_a$ spatiotemporal dynamics in complex urban environments.

# 1 Introduction

Air temperature ($T_a$) is a crucial meteorological variable that profoundly affects various facets of human welfare (Mora et al., 2017; Yuan et al., 2025; Zhang et al., 2023), including health (Tuholske et al., 2021), energy consumption (Perera et al., 2020; Wang et al., 2023a), and carbon emission (Li et al., 2024), to name a few. Its significance is particularly pronounced in urban areas, where 55% of the global population resides (UN Statistics Division, 2023). Rapid urbanization, characterized by extensive modifications in land use and land cover, has significantly altered the surface energy balance and the overlying climate (Arnfield, 2003; Oke et al., 2017). These transformations, in conjunction with the spatial heterogeneity of the built environment, anthropogenic activities and local landscapes, generate highly localized variations in $T_a$ at scales of approximately 100–1000 meters (Stewart and Oke, 2012). The increasing frequency and intensity of anomalous events under changing climate further complicates $T_a$ pattern in urban areas (Gao et al., 2024; Li and Bou-Zeid, 2013). Accurate and timely local-scale $T_a$ forecasting within cities presents great challenges, despite its critical role in urban management systems (Chen et al., 2024). The conventional approach to $T_a$ forecasting primarily relies on numerical weather prediction (NWP) models, which necessitate solving complex governing equations. However, generating high-resolution forecasts using this physics-based approach presents unique challenges due to urban characteristics, scales issues, and computational demand. First, existing NWP models often lack adequate parameterization schemes to represent complex processes within urban environments (Chen et al., 2011; Nogueira et al., 2022; Sharma et al., 2021). The requirement to specify numerous parameters for urban modules also introduces additional data challenges and uncertainties, which hinder their effective implementation (Chen et al., 2011). Second, substantial knowledge gaps persist in convective scale (<5 km resolution) modelling, including the absence of basic dynamical balances under nonhydrostatic formulations and the inherent complexity of resolving turbulent processes (Kendon et al., 2021; Schär et al., 2020; Yano et al., 2018). Third, the high computational demand of running NWP models, particularly when applying ensemble approaches to address forecast uncertainty, impede their feasibility for real-time operational use. These limitations constrain accurate local $T_a$ forecasts within cities.

Deep learning (DL) has emerged as a promising alternative approach for meteorological variables forecasting. These DL models can be primarily grouped into two paradigms: training with products of physics-based models or direct weather observations. The former paradigm typically relies on ECMWF's ERA5 reanalysis datasets to learn relationships between atmospheric states across successive time steps, and has recently achieved overall superior performance to state-of-the-art operational NWP systems (Bi et al., 2023; Lam et al., 2023; Price et al., 2024). However, this modeling paradigm inevitably inherits issues in urban areas, as the models are trained on data with insufficient urban representations and coarse spatial resolution. The latter paradigm utilizes in situ observations from weather stations or sensors and thus enables models to learn from data that authentically reflect local meteorological conditions (Effrosynidis et al., 2023; Wang et al., 2023b). The typical modeling approach adopts a time-series regression framework, wherein sequences of measurements at each individual locations are used to predict their respective values at subsequent time steps (Haque et al., 2021; Salcedo-Sanz et al., 2016; Wang et al., 2024; Yu et al., 2021). However, the forecast accuracy under this framework remains limited and has improved

only marginally despite the progressive adoption of increasingly sophisticated DL methods (Elsayed et al., 2021; Wang et al., 2024; Zeng et al., 2022). These limitations may stem from modeling approaches that rely purely on local time-series information, which on the one hand may fail to capture essential spatial contextual information, rendering the learning task underdetermined and semantically ambiguous (Iakovlev & Lähdesmäki, 2024). On the other hand, this inherently overlooks critical interactions with the surrounding environment that may be essential for accurate forecasting. Modeling observational data within cities provides a solution to deliver local-scale $T_a$ forecasts, while its potential remains underexplored.

With the development of graph neural networks (GNNs), which are capable of modeling discrete and irregularly distributed observation sites, pioneering studies have explored their use in connecting observations across locations to leverage spatial information for enhancing meteorological variable forecasting. Most existing efforts have focused on modeling large-scale observational networks sparsely distributed across broad regions, with the primary rationale being to address: 1) the atmospheric transport and advection processes among locations (Wang et al., 2020; Zhou et al., 2022); 2) weather propagation patterns (Wu et al., 2023); and (3) identify certain causal relationships among different cities (Li et al., 2023). Despite advances in understanding and modeling large-scale dynamics and their associated spatial interactions, it remains largely unknown whether observational network modeling approaches (i.e. incorporating spatial information) are effective at smaller intra-city scales. Furthermore, the underlying mechanisms and spatial dependencies that drive performance improvements in such scale remain unclear.

To study potential interactions among intra-city observations, we implement two GNNs with distinct spatial information aggregation mechanisms (directed and undirected) for short-term (1-6 hours) $T_a$ forecasting, using local measurements of $T_a$ and wind vectors across 16 locations in Hong Kong (Fig. 1a). In support of these GNN's implementation, we propose a novel framework Mix-n-Scale, which integrates optimization and ensemble processes to address the challenge in configuring graph topologies, particularly when prior knowledge of intra-city scale interactions is limited. Furthermore, we quantify the spatial information impacts on each location based on the GNN's information passing principle and compare the results with conventional time-series models where each location is modeled independently. This allows us to separate the contribution of intra-city spatial information on model behavior and understand the underlying mechanisms. This study offers critical insights into effective frameworks for modeling local observational data and sensor networks, which is increasingly important as crowd-sourced weather sensors continue to proliferate within urban environments (Chapman and Bell, 2018). The flexibility of this framework also makes it well-suited for adaptation to the modeling of similar environmental variables.

This paper is organized as follows. Section 2 provides details of datasets, problem formulation, DL models and their training framework, and metrics used in this study. In section 3.1, we first present the overall spatial characteristics of intra-city $T_a$. Section 3.2 presents modeling results for overall performance and extreme values, followed by an analysis disentangling the impact of spatial information on forecasting in Section 3.3. The spatiotemporal dynamics of $T_a$ forecast performance are further analyzed in Section 3.4. Section 4 presents the summary and conclusions.

## 2 Data and Methods

### 2.1 Datasets

Hong Kong, a densely populated coastal city at the southern edge of East Asia, features complex atmospheric circulation patterns due to its hilly terrain, land-sea contrasts, and heterogeneous urban morphology. make Hong Kong an ideal setting to examine a model's ability to capture local heterogeneity and intra-city $T_a$ dynamics. In this study, we use hourly meteorological data from 16 weather stations (Fig. 1a) operated by the Hong Kong Observatory. Although more stations exist, we limit our selection to sites with both $T_a$ and wind observations to ensure complete records for exploring the potential effects of wind. More specifically, three types of variables are incorporated into the model training. The first type includes local, spatially varying observations, including $T_a$ and wind speed (both U and V components). The second type includes globally uniform predictors across all sites, such as solar radiation (direct and diffuse) and mean sea level pressure; details and statistics of these variables are provided in Table 1. Additionally, we include spatial and temporal stamps for each site to represent its spatiotemporal context, but we do not incorporate detailed land use or urban morphology data, as the focus of this study is time-series forecasting at fixed observation sites rather than spatial prediction at unmeasured locations. Static variables are useful for spatial generalization, but their benefit is limited here, as the model focuses solely on short-range forecasts at the existing observation sites. It is worth noting that the final model performance is reported based on training without global predictors, as their inclusion did not yield improvement. These variables are retained only for ablation analysis (Section 3.2; Fig. 4b) to illustrate their potential influence on model performance. The entire dataset is divided into three disjoint subsets for training, validation, and testing. The training set covers four full years from 2016 to 2019, while the validation and test sets use data from 2020 and 2021, respectively, for model tuning and final performance evaluation.

**Table 1**

**Statistics of Variables Used for Model Training and Evaluation**

| Type | Input variable | Range | Mean | Unit | Abbreviation |
|---|---|---|---|---|---|
| **Global** | Direct solar radiation | [0, 3.64] | 0.38 | MJ/m$^2$ | - |
| | Diffuse solar radiation | [0, 2.24] | 0.31 | MJ/m$^2$ | - |
| | Mean sea level pressure | [977.8, 1037.3] | 1013.0 | hPa | - |
| **Local** | Zonal wind speed* | [-13.7, 7.3] | -0.2 | m/s | U |
| | Meridional wind speed* | [-13.2, 5.5] | -0.9 | m/s | V |
| | 2-m Air temperature | [-0.9, 38.2] | 23.4 | °C | $T_a$ |
| **Temporal** | Hour of day | [0, 23] | - | - | - |
| | Day of year | [1, 366] | - | - | - |
| | Month | [1, 12] | - | - | - |

| Spatial | Longitude | [113.92, 114.42] | 114.156 | degree | Lon |
| | Latitude | [22.20, 22.55] | 22.529 | degree | Lat |
| | Altitude | [4, 955] | 120 | m | Alt |

Note. *Positive value of U and V denote the wind is from the west and south, respectively.


## 2.2 Problem formulation and DL models

The task of $T_a$ forecasting at multiple locations is framed as a spatiotemporal prediction problem that uses existing observations to estimate the state of each location over several subsequent time steps. This is processed through a two-stage modeling approach. First, we embed the temporal dynamics at each location separately using Long Short-Term Memory (LSTM)

networks, which are effective for encoding temporal information of time-series (Greff et al., 2017). We also employ LSTM combining with decoder as a benchmark for time-series modeling using purely local information (Fig. 1b). Based on the time-series embeddings for each location, we then use GNNs to aggregate spatial information from irregularly distributed neighboring locations (Fig. 1a and c). The forecast horizon is set to six hours in this study, as longer lead times would require capturing large-scale dynamics that fall outside the scope of our target domain. The details of these two stages are as follows:

Temporal dynamics embedding: Let the input at a historical time step t as $X_t \in \mathbb{R}^{N \times F}$, where N represents the number of nodes (i.e., weather stations) and F denotes the number of predictor features. The LSTM captures the temporal evolution by processing observations over the previous $T$ time steps (time lag), yielding a set of temporal embeddings $\boldsymbol{h} = \{h_1, h_2, ..., h_N\}$, with each $h_i \in \mathbb{R}^{F'}$ where $F'$ is the dimensionality of the temporal embedding for nodes from 1 to $u$ (Hochreiter and Schmidhuber, 1997). This can be conceptually denoted as:

$$f_{LSTM}(X_t, X_{t+1}, ..., X_{t+T}) = \boldsymbol{h}. \tag{1}$$

Spatial information aggregating: Let the spatial connections between weather stations as a graph $\mathcal{G}(\mathcal{V}, \mathcal{E})$, with $\mathcal{V}$ is the set of nodes with their respective temporal embeddings $h_i$ as node features, and $\mathcal{E}$ is the edge denotes the connection between the nodes. Each node $i \in \mathcal{V}$ aggregates the representations from its immediate neighbors, $\{h_u^k, \forall u \in \mathcal{N}(v)\}$, into a single vector $h_{\mathcal{N}(i)}^{k-1}$. The k is the iteration of spatial aggregation (i.e., the depth of the GNN), and $k = 0$ corresponds to the initial embeddings

$\boldsymbol{h}$ from the LSTM. We implemented two GNN architectures, GraphSAGE (GSAGE; Hamilton et al., 2017) and graph attention network (GAT; Brody et al., 2021), because they representing two distinct learning mechanism for spatial information. GSAGE adopts an undirected graph structure with uniform neighbor weighting via mean aggregation, which was selected over max/min pooling in preliminary testing. In contrast, GAT learns directional influences by implementing an asymmetric attention mechanism that dynamically computes neighbor weights (Brody et al., 2021; Veličković et al., 2018), potentially capturing

directional influences and causal relationships where one node impacts another asymmetrically. However, GAT's greater modeling flexibility does not necessarily translate to superior performance. The details of two models are described formally in Text 1 in the Supplement.

## 2.3 Mix-n-Scale framework

Although GNNs offer a flexible modeling paradigm for integrating discrete local observations, determining appropriate graph
structure remains an open and challenging problem. Specifically, defining appropriate connectivity patterns between locations and selecting the optimal number of neighboring nodes represents a significant challenge. Such graph topologies are typically constructed through trial and error, involving extensive manual experimentation and iterative testing (Chen and Wu, 2022; Ma et al., 2023; Zheng et al., 2024).

This study therefore treats graph formation, along with time lag T, as hyperparameters and uses a greedy sequential method
to search for and optimize their optimal configuration. Moreover, one novelty of our approach is that we do not simply use the best-tuned model but additionally employ an ensemble-based approach to combine the top 10% of validated models composed of different graph topologies and time lags. We call this training framework Mix-n-Scale, and we refer to the trained model as a "hyper-model." To the best of our knowledge, such an ensemble-based approach using various graph structures for sensor network modeling has not been studied or examined. Our rationale for employing this framework is twofold: (1) the selection
of neighboring stations to establish connections and the time-series length potentially incorporates information from different spatiotemporal scales, enriching the representation of existing information; (2) since DL model training accounts for the majority of computational resources in model development process (conventional trial and error or our optimization process), while each inference (i.e., forecast) can be completed within seconds with minimal computational cost compared to the training stage (Goodfellow et al., 2016), our proposed hyper-model approach incurs marginal additional computational overhead in
real-world applications while more effectively leveraging the substantial resources already required for model development. Specifically, we use tree-structured Parzen estimator (Bergstra et al., 2011) based on the its loss on validation set, examining various edge formation strategies (from self-connection to connection across all neighbors) for the graphs, look-back lengths (from 1 to 200 time steps) for the input time-series, and varying model architecture hyperparameters. The selection process can be formulated as follows:

$$\hat{\boldsymbol{\theta}}(\boldsymbol{\lambda}) \in \operatorname{argmin} \mathbb{E}_{(\mathbf{x},y)\in\mathcal{D}}\big[\ell\big(f_\theta(\mathbf{X}, y, \boldsymbol{\theta}, \boldsymbol{\lambda})\big)\big], \tag{2}$$

where $\ell$ represents the mean squared error loss. X and y denote individual features and labels, respectively, that comprise the dataset $\mathcal{D}$. $f_\theta$ represents corresponding test DL architecture, where $\boldsymbol{\theta}$ encompasses all the model trainable parameters; $\boldsymbol{\lambda}$ represents hyperparameters determining the graph structure, time lags and a few learning hyperparameters including learning rate and hidden dimensions. $\mathbb{E}_{(\mathbf{x},y)\in\mathcal{D}}[\cdot]$ stands for the expectation with the distribution over $\mathcal{D}$. The search process iterates 100
times and selects the model based on the top 10% (10 out of 100) $\boldsymbol{\lambda}$ hyperparameter settings.

## 2.4 Metrics

### 2.4.1 Temperature variability metrics

The daily $T_a$ evolution pattern can be primarily described by two metrics, including the mean daily value and the magnitude of diurnal variation. In this study, we introduce diurnal temperature standard deviation (DTSD) to quantify and characterize

the intensity of diurnal $T_a$ fluctuations at each location, serving as an indicator to show local $T_a$ pattern. For location i, the DTSD is defined as:

$$\text{DTSD}_{(i)} = \sqrt{\frac{1}{24D}\sum_{j=1}^{D}\sum_{h=1}^{24}\left(T_{a(i,j,h)} - \bar{T}_{a(i,j)}\right)^2}, \tag{3}$$

where $T_{a(i,j,h)}$ is the $T_a$ at location I , on day j, at hour h; $\bar{T}_{a(i,j)}$ is the mean daily $T_a$ at location I on day j; D is the Total number of days in the datasets.

### 2.4.2 Model evaluation metrics

We calculate the root mean squared error (RMSE) and Bias to evaluate model performance. These metrics are calculated as follows:

$$\text{RMSE} = \sqrt{\frac{1}{NT_h}\sum_{t=1}^{T_h}\sum_{i=1}^{N}\left(\hat{T}_{a(i,t)} - T_{a(i,t)}\right)^2}, \tag{4}$$

$$\text{Bias} = \frac{1}{NT_h}\sum_{t=1}^{T_h}\sum_{i=1}^{N}\left(\hat{T}_{a(i,t)} - T_{a(i,t)}\right), \tag{5}$$

where $\hat{T}_{a(i,j)}$ is the predicted $T_a$ at location i at time t. $T_{a(i,t)}$ is corresponding true $T_a$. N is the total number of the locations, and $T_h$ is the total number of hourly samples. Here, a positive bias indicates overestimation, and vice versa for a negative bias.

### 2.4.3 Local oscillation index (LOI)

LOI is a metric that we proposed based on the graph Laplacian (Hamilton et al., 2017) that quantifies the surrounding information inflow to each node. This is utilized to quantify the impact of spatial information from surrounding nodes on local forecasting. Mathematically, let $T_{a(i,t)}$ be the $T_a$ observed at location $i$ at the time $t$, and $T_{a_{\mathcal{N}(i,t)}}$ as the mean $T_a$ of the neighboring stations of station $i$ at the same time is calculated as:

$$T_{a_{\mathcal{N}(i,t)}} = \frac{1}{N}\sum_{u\in\mathcal{N}(i)}T_{a_{u,t}}, \tag{6}$$

where $\mathcal{N}(i)$ represents the set of neighboring stations to $i$, and $N$ is the number of neighbors. For each station $i$, the deviation of $T_a$ from its neighbors at any given time t is $\Delta T_{a(i,t)}$:

$$\Delta T_{a(i,t)} = T_{a(i,t)} - T_{a_{\mathcal{N}(i,t)}}. \tag{7}$$

Based on $\Delta T_{a(i,t)}$, one can calculate the historical normal deviation of one station from its neighbors by averaging the deviations over records across the training period. The historical normal deviations $\overline{\Delta T}_{a(i,h,m)}$ for location $i$ at hour $h$ and month $m$ is calculated as follows:

$$\overline{\Delta T}_{a(i,h,m)} = \frac{1}{|T_{h,m}|}\sum_{t\in T_{h,m}}\Delta T_{a(i,t)}, \tag{8}$$

where $T_{h,m}$ represents the set of all historical time points corresponding to hour $h$ and month $m$, and $|T_{h,m}|$ is the number of time points. And then the LOI is calculated as follows:

$$LOI_{i,t} = \Delta T_{a_{(i,t)}} - \overline{\Delta T}_{a_{(i,h,m)}}. \tag{9}$$

LOI essentially reflects how a node differs from its surroundings while eliminating climatological differences. This primarily captures the effect of the graph processing procedure and helps disentangle the impact of spatial information. Note that LOI is an hourly metric, rather than reflecting daily deviation.

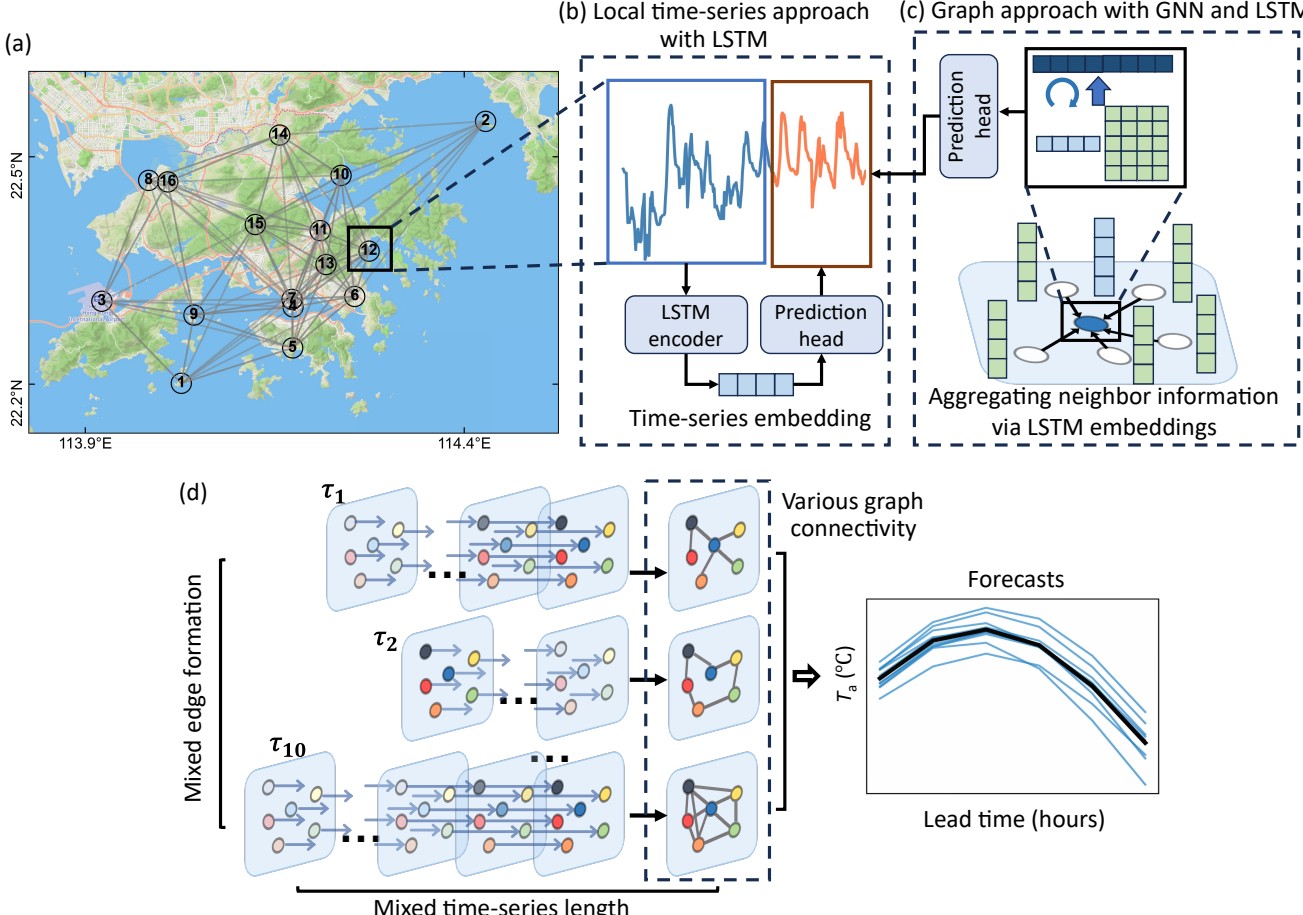

**Figure 1.** Schematic of the modelling framework. **(a)** Spatial distribution of weather observation stations across Hong Kong (basemap © Mapbox), with location IDs labeled. The edges between stations represent the schematic GNN structure, showing nine connections per node. **(b, c)** Conceptual diagram comparing the local time-series modeling approach with the graph-based approach, in which LSTM-based temporal embeddings are spatially aggregated using GNN across neighboring stations. **(d)** Overview of the Mix-n-Scale framework, which leverages intra-city observations using diversely configured GNNs.

## 3 Results and Discussions

### 3.1 Intra-city $T_a$ characteristics

We first present the intra-city spatiotemporal dynamics of $T_a$ within our study areas. Overall, the mean $T_a$ patterns is relatively homogeneous, with majority sites recording mean values within a narrow range of 23.2°C to 24.2°C. Two notable exceptions are high elevation sites, location 15 (elevation: 955 m) and location 13 (elevation: 572 m), which exhibit the lowest annual mean $T_a$ of 17.6°C and 19.6°C, respectively. In contrast, Hong Kong International Airport, location 3, dominated by concrete structures with high thermal inertia, records the highest mean $T_a$ of 24.8°C.

In comparison, diurnal $T_a$ fluctuation exhibits a more heterogenous pattern. The DTSD (Section 2.4.1) evenly distributed from 1.3°C to 2.5°C, indicating substantial relative spatial variability (Fig. 2b). The lowest DTSD of 1.3°C occurs at the mountain peak (location 15), while the highest value of 2.5°C is observed at location 14 in the northern inland suburban area. Notably, diurnal fluctuations tend to be greater in northern areas at shown in the right panel of Fig. 2b, likely due to reduced oceanic thermal moderation and stronger influence from continental air masses (Scheitlin, 2013). The relative magnitude of variation among locations reveals similar mean value patterns but more pronounced differences in diurnal fluctuations.

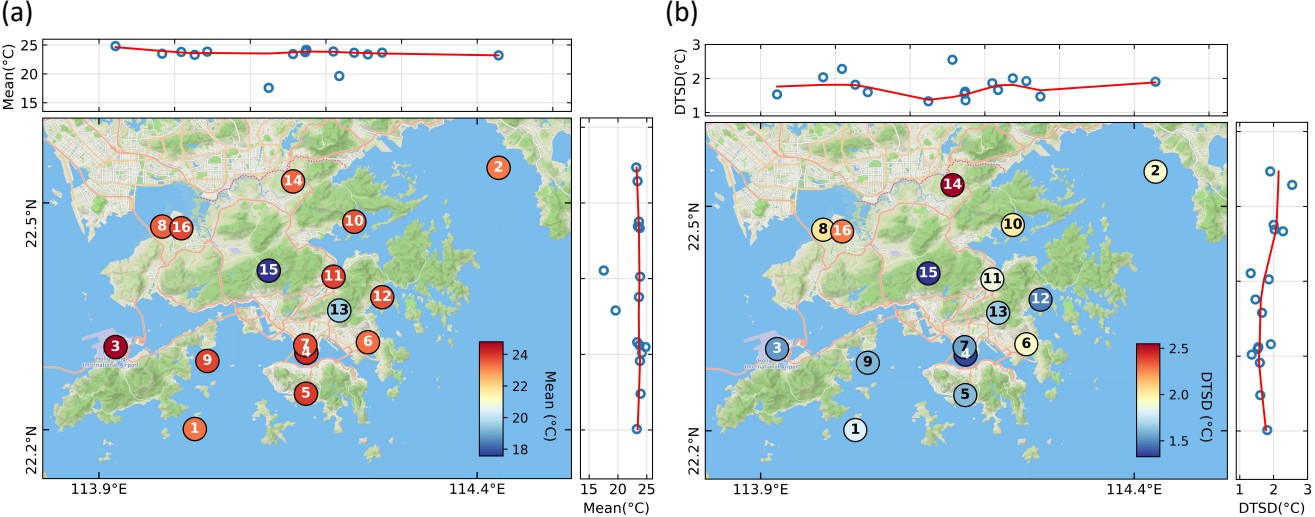

**Figure 2.** Spatial distribution of **(a)** mean $T_a$ and **(b)** mean diurnal standard deviation (DTSD) over the six-year datasets (basemap © Mapbox). Node colors indicate the magnitude at each location, and numbers denote location IDs using different colors for clearer visualization. The upper and right panels show corresponding values along longitude and latitude, respectively, with the solid line indicating a LOESS-smoothed value.

### 3.2 Overall evaluation of DL models

We evaluate the DL models based on their average performance for 1–6 hour forecasts across 16 weather stations in Hong Kong. The graph-based models consistently outperform purely local time-series models. Specifically, GSAGE achieves the

lowest RMSE of 0.96°C, followed by the GAT with 1.03°C, both outperforming the LSTM baseline (1.06°C). These results
highlight the benefit of incorporating spatial information from neighboring stations for local $T_a$ forecasting.

Our Mix-n-Scale framework achieves varying performance gains across different DL models (red triangles, Fig. 3a). Since
simple LSTM does not involve graph structure, we therefore apply a naïve hyperparameter ensemble that includes models with
varying learning rate and hidden dimensions and time lags. While hyper-LSTM shows only marginal gains over the single
LSTM (Fig. 3a), applying the Mix-n-Scale framework to GSAGE yields roughly threefold greater improvements, highlighting
its suitability for our graph-based task. Overall, Hyper-GSAGE reduces RMSE from 1.06 °C to 0.92 °C, representing a 12.5%
enhancement over the best LSTM. Its performance remains highly stable across different ensemble sizes, achieving optimal
accuracy when incorporating the top ~10% of models from validated pool (10 out of 100; Fig. S1 in the Supplement). Building
such a pool typically requires around 50 hyperparameter trials drawn from a broad initial search space (Fig. S2 in the
Supplement), which can be completed within 10 hours on a single RTX 4090 GPU. Once finalized, the model generates
forecasts within seconds, enabling efficient real-time applications.

Across all forecast horizons, Hyper-GSAGE consistently outperformed the baseline models, exhibiting reduced errors and
uncertainties. Moreover, compared with recent large-scale evaluations of multiple physics-based (e.g., ECMWF, GFS) and
data-driven models (e.g., Pangu, Fuxi, Fengwu) against observations from over 2,000 stations across China, which reported
near-surface temperature forecast errors typically exceeding 2 °C at a 3-hour lead time (Xu et al., 2025). In contrast, Hyper-
GSAGE achieve RMSEs ranging from 0.88–1.17 °C for 3–6 hour forecasts, demonstrating a clear advantage in local-scale $T_a$
forecasting skill.

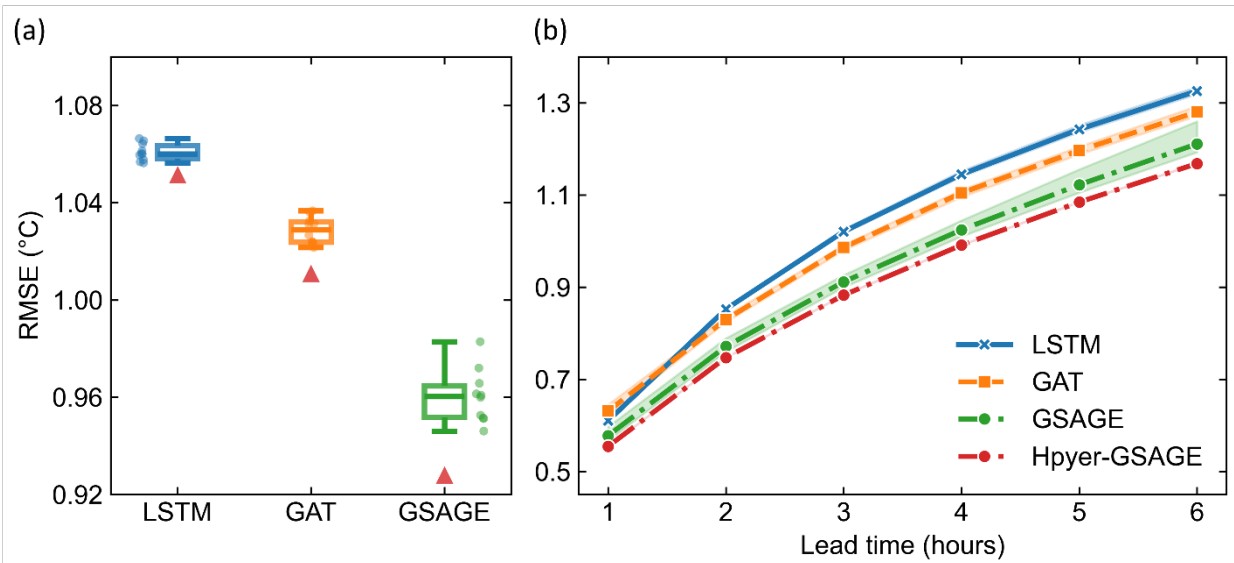

**Figure 3.** $T_a$ forecast accuracy for the next 6 hours over 16 studied weather stations by different models. **(a)** Overall results of three deep
learning models. Each box contains the 10% best individual models (10 out of 100 trained models based on validation results). Box plots
show the median (line), 25–75% range (box), and whiskers are drawn to the farthest datapoint within 1.5 inner quantile range. The red

triangle denotes the model accuracy with the Mix-n-Scale framework based on the 10% best models. **(b)** Forecast accuracy at different lead times. Shaded areas denote the range of RMSEs among the 10% best models.

Between the two spatial information learning approaches, it is worth noting that GSAGE's simple mean aggregation of neighbor's temporal embeddings outperforms GAT's adaptive attention mechanism, which assigns dynamic weights to neighbors. Although GAT theoretically offers greater flexibility by identifying variable inter-station relationships, and wind vectors are included to provide potential directional cues, this advantage does not manifest here and may instead lead to overfitting issues. This suggests that, at the intra-city scale, there may be no distinct "upstream" information flow or dominant "super-nodes," or, if present, such relationships may occur at shorter timescales. We further examined this hypothesis from a statistical perspective by conducting cross-correlation analyses among station observations with varying time lags to assess whether $T_a$ changes at one site could precede others. The lag-shifted results show that most inter-station correlations peak at zero lag (even for the most distant station pair), with only a few pairs exhibiting slightly higher correlations at ±1 h (Fig. S3 in the Supplement).

Based on the GSAGE model, we further examine how graph construction affects model performance. We find that connecting each station to its nearest neighbor generally yields better performance than linking to the most distant ones (Fig. 4a), even though the latter could capture broader meteorological context and longer-range propagation. This result is consistent with our cross-correlation analysis. Regarding edge formation, the GSAGE model tends to perform better when each node connects more neighbors, particularly larger than nine (Fig. 4a). Another key structural factor is graph depth, which determines how many hops of neighbor information each node can access. We observe more than a 5% RMSE reduction when using two GNN layers compared to a single layer (Fig. S4 in the Supplement). Although a single layer with full connectivity can theoretically access the entire graph, adding a second layer does not expand the receptive field but introduces additional nonlinearity and feature-transformation capacity, potentially improving model expressiveness. However, deeper architectures that repeatedly aggregate neighbor information do not provide further gains in our case and may instead lead to over-smoothing, making node representations less distinguishable. Collectively, these findings clarify the optimal GNN configuration for the $T_a$ forecasting task and indicate that domain-wide spatial context is likely to play a key role in enhancing model performance, which will be further explored in Section 3.3. The remaining key hyperparameters and their optimal ranges are summarized in Text S2 in the Supplement.

To understand the significance of each predictor, we perform ablation experiments by systematically removing predictors. Including wind vectors reduces RMSEs from 0.98 to 0.96 °C, whereas global variables that are uniform across stations (i.e. solar radiation and MSLP) do not further enhance forecast accuracy (Fig. 4b). This is likely because such variables are more physically meaningful when their spatial patterns and gradients are represented (e.g., pressure gradients that drive large-scale flows or synoptic features such as troughs and ridges). When incorporated as single-point values, they provide limited information and may even introduce noise.

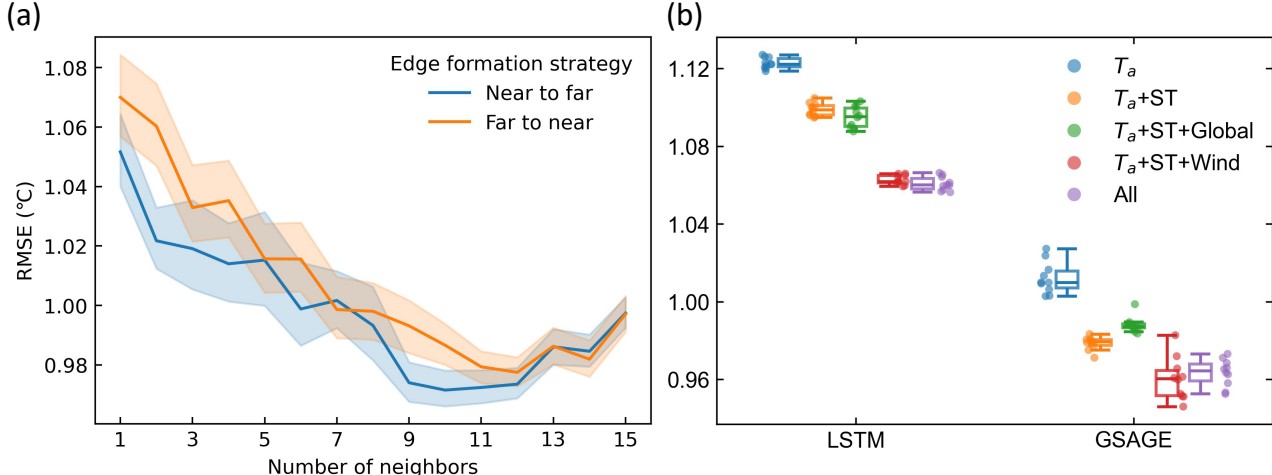

Figure 4. (a) Variation of RMSE with the number of neighboring nodes used to form edge connection under two strategies. The solid curve denotes the mean RMSE, and the shaded area represents the standard deviation across models trained with different hyperparameter settings. (b) Impact of predictors on $T_a$ forecast accuracy. Performance of LSTM and GSAGE models with various combinations of predictors. ST represent spatial and temporal predictors. Global represents global meteorological predictors (uniform across stations) include direct and

290 diffuse solar radiation, and mean sea level pressure. Wind includes zonal and meridional wind speed at individual stations. Detailed variable descriptions are available in Table 1. Box plots show the median (line), 25–75% range (box) based on the 10% best models, and whiskers are drawn to the farthest datapoint within 1.5 inner quantile range.

Does Hyper-GSAGE preserve extreme values? Given that the model is essentially generated through a multi-model

ensemble approach, a major concern is that the results tend to smooth predicted values and sacrifice the ability to capture extreme values (Knutti et al., 2010; Wilks, 2011). Therefore, we examine the distribution of the 5% most extreme values (both warmer and colder) in model forecasts. We find that predicting these values is highly challenging for all models, where we observe rightward-shifted forecasts for colder values and more pronounced leftward shifts for warmer values, reflecting overestimation of low and underestimation of high $T_a$ (Fig. 5a). The greater cold bias for warmer values indicates inherent

challenges in capturing extreme high temperatures. However, it is worth noting that Hyper-GSAGE demonstrates better alignment with distribution of observations.

Furthermore, we compare model accuracy under extreme conditions using the predicted and corresponding observed values (Fig. 5b). For colder values, both GSAGE and Hyper-GSAGE reach comparable results, significantly outperform than LSTM model by reducing RMSE from 1.76°C to ~ 1.50 °C. However, for warmer values, while GSAGE improved RMSE from

305 1.62°C to 1.53°C, Hyper-GSAGE achieves clear better results (RMSE: 1.41°C) with additional bias reduction from -1.13°C to -0.99°C. These results demonstrate that Hyper-GSAGE enhances performance under both overall and extreme conditions.

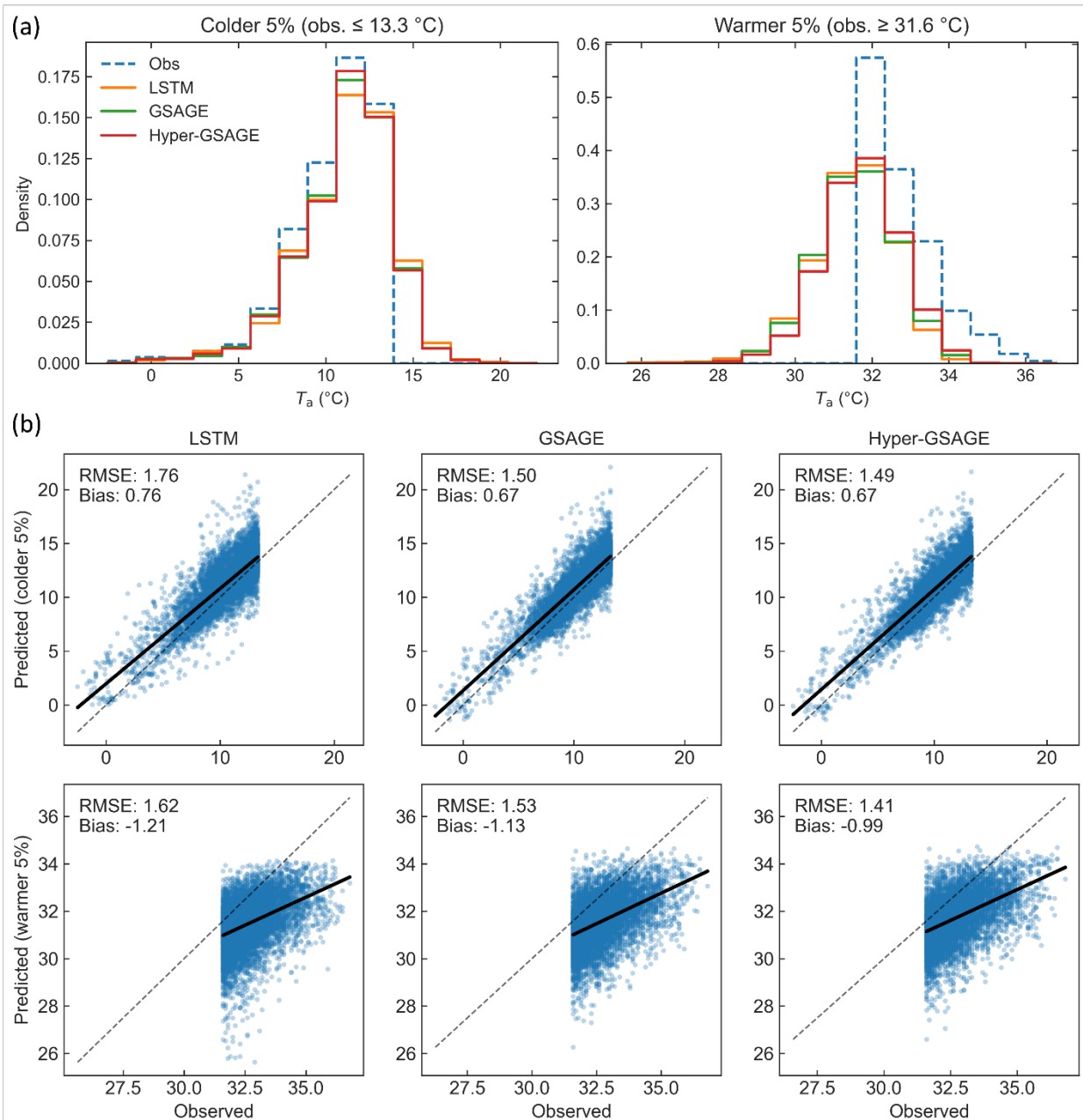

**Figure 5. (a)** Probability density distributions of the observed $T_a$ and corresponding predictions (6-hour lead time) from the LSTM, GSAGE, and Hyper-GSAGE models for the coldest 5% (obs. ≤ 13.3 °C, left) and warmest 5% (obs. ≥ 31.6 °C, right) of samples. The dashed blue line represents the observed distribution, while solid lines show predictions from each model. **(b)** Scatter plots comparing observed and predicted $T_a$ (6-hour lead time) for the same extremes (coldest 5%, top row; warmest 5%, bottom row). Each column corresponds to a different model: LSTM (left), GSAGE (middle), and Hyper-GSAGE (right). The 1:1 line (dashed) indicates perfect prediction; solid black lines show the linearly fitted regression trend for each case. RMSE and Bias are provided to quantify model performance for the respective extremes.

### 3.3 Impacts of intra-city scale spatial information

The superior performance of graph-based models demonstrates the critical influence of spatial information, motivating investigation of the underlying mechanisms driving these improvements. This requires quantifying both spatial information inflow to each node and how model behavior changes after incorporating this information. The latter is relatively straightforward to identify by directly calculating the difference between predictions from graph-based models and local time-series-based LSTM models. However, quantifying spatial information flows to individual nodes is challenging because these flows are learned as high-dimensional latent representations in an end-to-end manner by DL models. To explicitly quantify this information, we propose LOI, an index calculated based on GSAGE's message-passing process (Section 2.4.2) that allows us to track how spatial information influences model behavior. In our context, LOI can be interpreted as the extent to which a location's $T_a$ anomaly deviates from the mean value of its neighboring nodes.

We observe an inverse relationship between the LOI and its impact on $T_a$ forecasts (Hyper-GSAGE minus LSTM, denote as $\Delta\hat{T}_a$ hereafter), as shown in Fig. 6a. This indicates that Hyper-GSAGE tends to adjust a node's prediction upward (positive $\Delta\hat{T}_a$) when its current $T_a$ value is abnormally below its neighbors (negative LOI), as illustrated in Fig. 6b. In other words, this promotes convergence of mean $T_a$ patterns across locations. The rationale behind is that daily mean $T_a$ maintains similar patterns within the city as noted in Section 3.1, with a limited variance of $0.35°C^2$ among locations (Fig. S5 in the Supplement). The spatially stable mean $T_a$ pattern therefore serves as a dynamic indicator that constrains and refine forecasts on each node's diurnal amplitude rather than relying solely on local time-series trajectories. Graph regularization naturally enforces such adjustment through its smoothness property (Kipf and Welling, 2017), enhancing model's capacity to modulating local heterogeneous response. We term this effect "mean state regularization" for $T_a$ forecasting. Fig. 5c presents a case study in location 14 that clearly demonstrates this effect during January 12th-15th when weather starts turning to fine condition (The Weather of January 2021 in Hong Kong, 2025), when $T_a$ pattern shifts to stronger fluctuation with higher cooling and heating rate. Since this location exhibits abnormally cooler $T_a$ than its neighbors during nighttime, Hyper-GSAGE produces additional upward adjustment in its subsequent daytime $T_a$ forecasts compared with LSTM, effectively capturing the dynamics, especially the daily peak $T_a$ across those days. In contrast, the LSTM forecasts largely replicate the time-series evolution from the preceding day (Fig. 5c).

In essence, LOI reflects the heterogeneous local Ta response that are jointly shaped by environmental factors and background weather conditions. A more direct investigation of how these modulate local $T_a$ diurnal variation amplitude, as well as how the performance gains of GSAGE vary across different cities, would be valuable directions for future work. It is also important to note that our current interpretation offers only a conceptual representation of the GSAGE model, as it cannot fully encapsulate the complexity of deep learning architectures involving multi-layer nonlinear propagation and higher-order feature interactions.

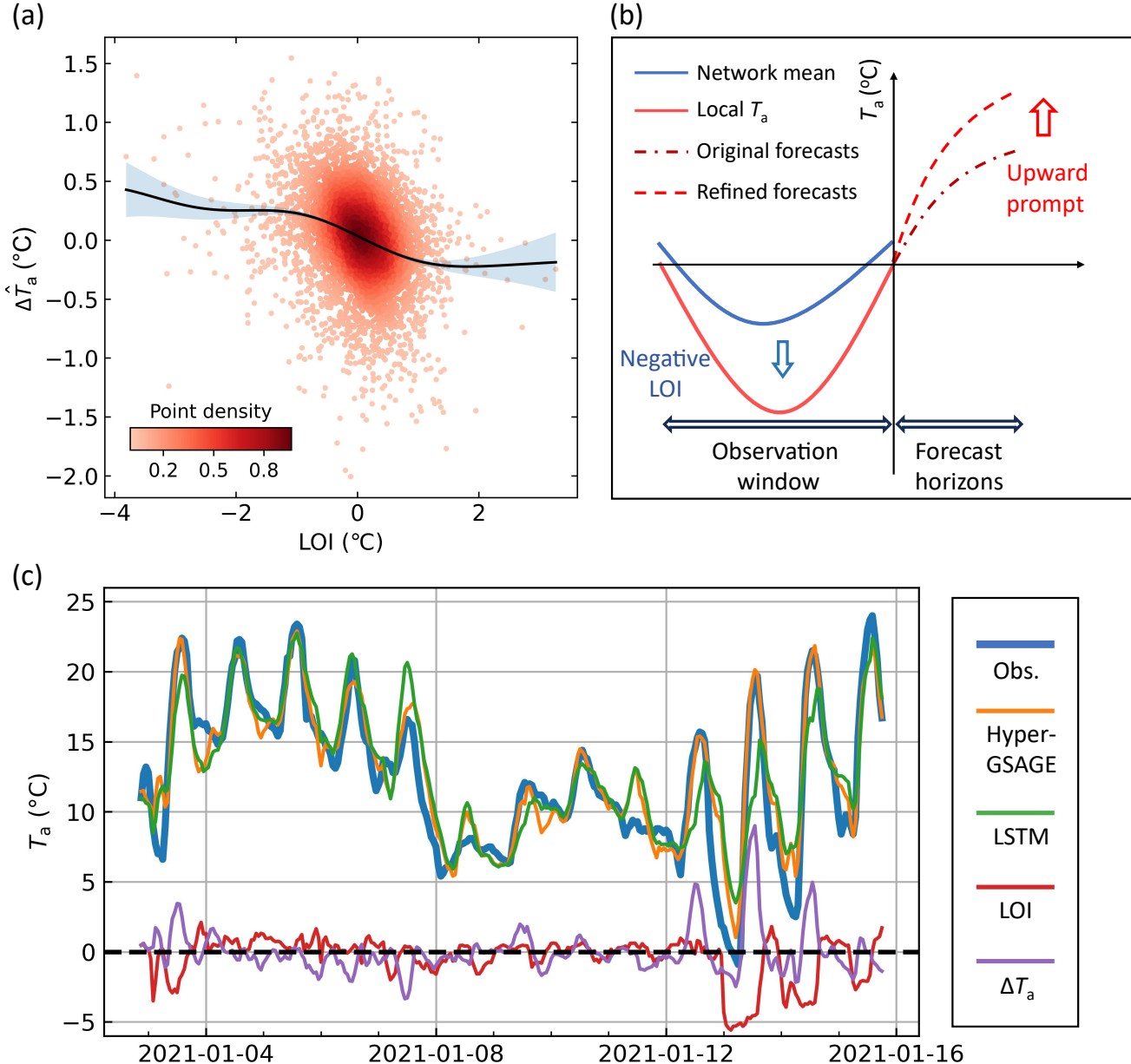

**Figure 6.** Change in $T_a$ forecast after incorporating spatial information with Hyper-GSAGE. **(a)** Negative relationship between LOI and $\Delta\widehat{T}_a$. The difference ($\Delta$) is defined as Hyper-GSAGE minus LSTM output at the lead time of 6-hour. Each point in the scatter denotes a daily mean value at a single station, with color indicating the point density. The trend is fitted by Gaussian process regression, with shaded areas denoting the 95% confidence interval of the probabilistic model. **(b)** A diagram illustrating how spatial information influences $T_a$ forecasting at one specific location, where a negative LOI prompts the model to forecast a higher $T_a$, thereby refining the local magnitude. **(c)** A case study illustrating the temporal evolution of observed $T_a$ and forecasts produced by the LSTM, Hyper-GSAGE models and their

difference ($\Delta T_a$) at a 6-hour lead time. The LOI evolution is shown relative to the forecast initialization time to reflect the information can be received by the model.

## 3.4 Spatiotemporal dynamics of forecast performance

Following the successful development of the Hyper-GSAGE model, we further evaluate its spatiotemporal forecast performance to elucidate the variability and underlying dynamics of prediction errors. The results reveal a pronounced diurnal contrast, with RMSEs increasing during the daytime and peaking between 10:00–14:00 (1.27–1.40 °C), coinciding
with the warmest period of the day (Fig. S6 in the Supplement). In contrast, nighttime forecasts, particularly between 00:00–04:00, exhibit the lowest RMSEs (0.61–0.63 °C). This pattern remains consistent even when RMSEs are normalized by the mean hourly $T_a$ of the corresponding periods (Fig. S7 in the Supplement). The distinct diurnal variation in forecast skill can be primarily attributed to differences in $T_a$ evolution dynamics between day and night. During daytime, solar radiation–induced surface heating and subsequent atmosphere–land interactions introduce strong perturbations, amplifying $T_a$
variability and increasing forecast difficulty. After sunset, however, $T_a$ evolves more smoothly under stable boundary-layer conditions, resulting in reduced variability and lower forecast errors. This diurnal contrast is further supported by the autocorrelation analysis (Fig. S8a in the Supplement), which indicates substantially higher nighttime persistence (~0.94) compared with daytime, particularly around 12:00–14:00 when persistence reaches a minimum (~0.84) at the 1-hour lag. A similar contrast is also observed for the 1-day lag (same hour on the previous day), with persistence values of ~0.75 at night
and ~0.57 during midday. Collectively, these results demonstrate that daytime $T_a$ variability is more dynamic and thus inherently less predictable from a statistical perspective. Seasonally, both summer and winter exhibit elevated forecast errors (RMSEs of 1.00 °C and 0.92 °C, respectively). While summer remains relatively stable under the control of the subtropical high-pressure system (24-h lag autocorrelation of 0.37 in summer, compared with −0.01 in winter; Fig. S8b in the Supplement), stronger radiative forcing and turbulent energy exchange within a more energetic atmosphere likely contribute
to greater short-term $T_a$ variability at hourly scales. This is reflected by the lower 1-hour autocorrelation (0.48 in summer compared with 0.72 in winter; Fig. S8b in the Supplement), presenting greater challenges for short-range forecasting.

Forecast accuracy shows substantial spatial variability, with RMSEs ranging from 0.72 to 1.10 °C (Fig. 6b). Two locations within the most densely developed urban areas show the lowest RMSEs (0.72 and 0.76°C for location 4 and 7, respectively). These locations are surrounded by high-rise buildings in the urban core, where local areas typically have large thermal inertial
and reduced ventilation. Location 3 records the highest mean Ta at the airport while demonstrating a relatively low RMSE of 0.82°C. The highest RMSEs are found at locations 14 and 10 where they are in the most inland place (expect the mountain station) and interface of sea water and freshwater reservoir with RMSE of 1.10 and 1.07°C.

Across all classified periods, we find that the heterogeneous spatial RMSEs within the city are highly positively correlated with corresponding observed local variability (Fig. 7c), as measured by the standard deviation (SD) $T_a$ observations at each

location. While we find this pattern varies dynamically among periods. Summer patterns exhibit distinct diurnal differences. During daytime, local $T_a$ variability diverges significantly across locations (SD from 1.6 to 3.0°C). We treat location 15 as a proxy for background weather conditions, as it is situated atop the city's highest mountain and is therefore minimally perturbed by atmosphere-land interactions. The mountain-top station shows the least local variability and forecast errors, which aligns with Hong Kong's stable summer weather patterns typically dominated by subtropical high-pressure systems. The remaining locations experience greater $T_a$ variability and associated forecast errors during daytime, likely caused by intense solar radiation and subsequent thermal instability and convective turbulence. This variability, along with associated RMSEs, diminishes and converges at night, highlighting the substantial uncertainties induced by solar radiation in generating $T_a$ instability and spatial heterogeneity during summer. Winter presents a different scenario. Although daytime still shows higher $T_a$ variability and RMSEs, a wide spread persists throughout the nighttime. This pattern likely reflects the influence of more variable synoptic conditions, particularly monsoon surges and cold-front passages, as evidenced by the markedly increased variability at the mountain-top station (location 15) during this period (Fig. 7b). In this context, local ventilation conditions (e.g., building configuration and urban morphology) and thermal properties play crucial roles in modulating how local thermal environments respond to background forcing. Indeed, we observe that stations with higher forecast errors typically under prevailing northerly winds during winter nights (Fig. S9 in the Supplement).

Despite the temporal variability, consistent spatial patterns emerge across periods. Location 10, situated at the interface between the sea and Hong Kong's largest freshwater reservoir, consistently exhibits the largest forecast errors during daytime in both seasons. This behavior can be attributed to pronounced thermal contrasts induced by strong solar radiation and the resulting complex local sea–lake–land breeze circulations and turbulence. In contrast, its nighttime $T_a$ remain relatively stable. At the most inland site (location 14), we observe persistently higher daytime $T_a$ and lower nighttime $T_a$ across both seasons (Fig. S10 in the Supplement). This pattern likely arises from reduced moderation by sea breezes and stronger advection of warm air towards the inland during the day, followed by more efficient cooling of land breezes at night. Such a amplification or diurnal contrast on the inland areas has also been documented in other coastal cities (Bauer, 2020; Yang et al., 2023). Conversely, the most densely urbanized areas display consistently lower $T_a$ variability and smaller forecast errors across all periods. This stable pattern can be explained by that dense high-rise structures tend to suppress daytime heating while enhancing nocturnal heat retention (Oke et al., 2017; Shi et al., 2024). Collectively, these complex spatiotemporal dynamics underscore the diverse physical processes governing intra-urban temperature variability and forecast uncertainty, highlighting the need for refined model representations and period-specific evaluations in urban $T_a$ prediction studies.

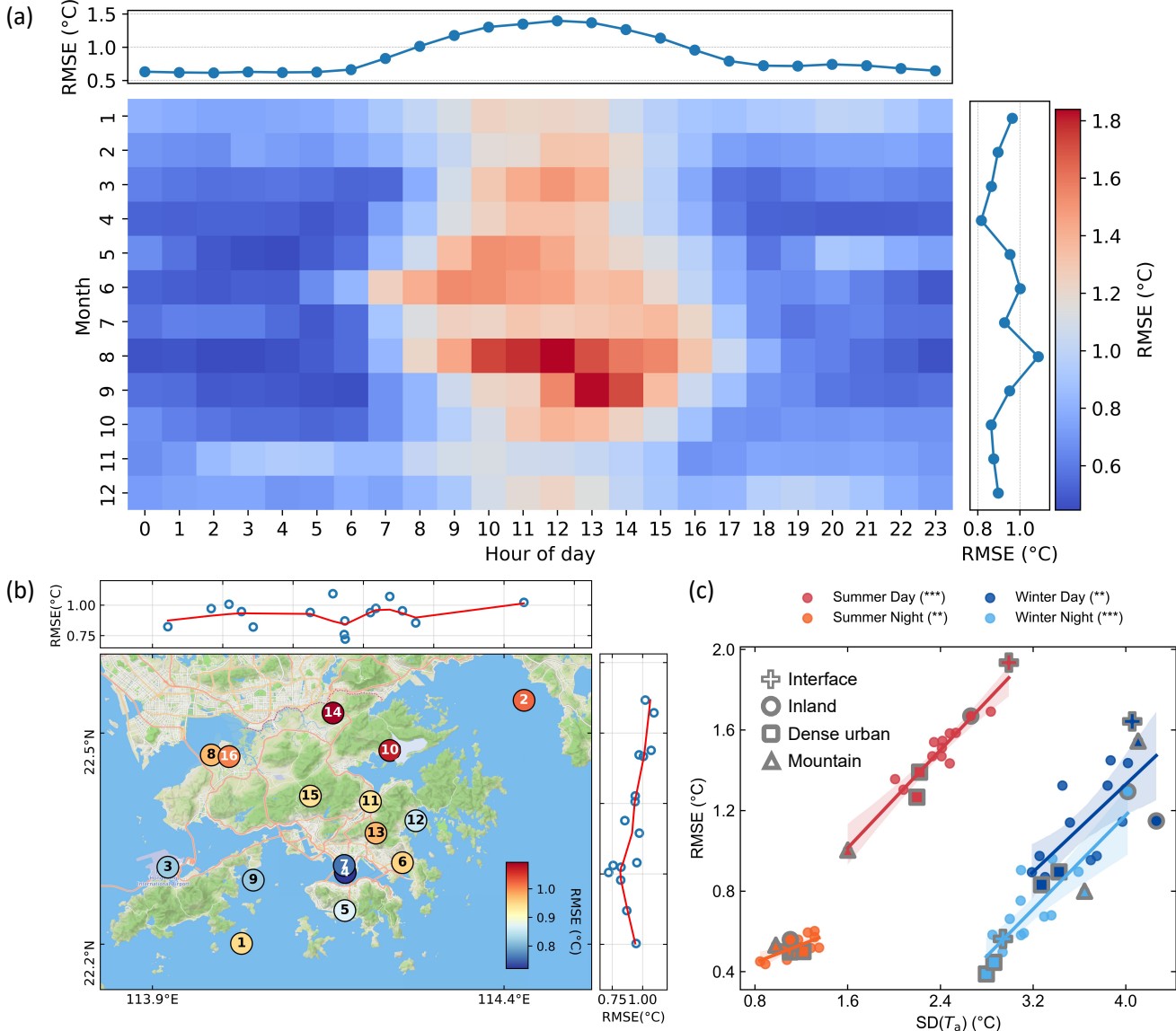

**Figure 7. (a)** Temporal variation of forecast accuracy. The top and right panels display the mean hourly RMSE of Hyper-GSAGE aggregated over hours and months, respectively. **(b)** Same as Fig. 2 but for Spatial distribution of RMSEs (basemap © Mapbox). The light-blue area to the right of Location 10 represents a freshwater reservoir. **(c)** The relationship between local $T_a$ variability and forecast RMSE. Each point represents a location, and shaded areas indicate 95% confidence intervals derived from bootstrapping. Key locations discussed in the text are highlighted using the shapes indicated in the legend. Three (***) and two (**) asterisks next to each period indicate that the relationship is significant at $p \leq 0.001$ and $p \leq 0.01$, respectively.


## 4 Concluding remarks

This study highlights the importance of a graph-based approach for modeling intra-city observation networks collectively to improve short-range $T_a$ forecasts at individual locations. We demonstrate that an undirected graph formation using the GSAGE model can refine local forecasting by effectively enforcing constraints captured from the mean states of neighboring observations, as revealed by our proposed LOI. Within the proposed Mix-n-Scale framework, the Hyper-GSAGE model produces more accurate forecasts under both general and extreme conditions, achieving an average RMSE reduction exceeding

12.5% for 1–6 hour forecasts compared with the conventional time-series method.

The spatial distribution of $T_a$ forecast accuracy exhibits substantial heterogeneity that strongly correlates with local $T_a$ variability, while these patterns vary considerably across temporal periods. Summer demonstrates distinct diurnal variations in spatial patterns, where daytime conditions substantially amplify both spatial heterogeneity and error magnitude, suggesting a critical role for solar radiation. In contrast, winter exhibits more consistent diurnal patterns, where local ventilation and

thermal properties start emerging as critical factors under a more variable background condition.

Given our focus on intra-city spatial interactions, our models are developed without incorporating meso-scale weather information. We acknowledge this design choice limits our ability to capture weather propagation beyond the domain boundaries, and we therefore constrain the forecast horizon to 6 hours in this study. Incorporating large-scale patterns, such as cold frontal passages propagating from outside the domain, through lateral boundary conditions or broader-scale atmospheric

predictors could be critical for capturing overall trends, particularly during the more variable winter season. Nonetheless, as $T_a$ patterns are influenced by various local circulations, integrating high-resolution computational fluid dynamics simulations holds great potential for elucidating intra-city airflow dynamics and further refining forecast accuracy through hybrid modeling approaches. Hyper-GSAGE serves as a foundational yet flexible framework for modeling local observation networks, with the capability to integrate this information with NWP systems or their DL-based surrogates, thereby leveraging advantages from

both physics-based and data-driven approaches. With the increasing deployment of IoT weather observation sensors in cities (Chapman and Bell, 2018), such models offer substantial potential for improving urban environmental management at finer spatiotemporal scales, providing a pathway toward more precise and intelligent oversight of urban systems.

## 4 Competing interests

The authors declare no conflicts of interest relevant to this study.


## 5 Code availability

The meteorological data for Hong Kong were obtained from the Hong Kong Observatory, which can be acquired from https://www.hko.gov.hk/en/cis/climat.htm. The workflow, model files, and outputs generated during testing and validation are publicly available on Zenodo (Wang, 2025) under the Creative Commons Attribution 4.0 International License.

## 455  6 Author contributions

The paper was designed and written by HW and JY. The model was developed by HW and subsequently discussed and validated with JT and JZ. HW, JY, JT and JZ performed the result analyses. All co-authors actively contributed to the extended discussions, refinement of the study design, and critical review of the final manuscript.

## 7 Acknowledgments

This work was supported by the National Natural Science Foundation of China for Excellent Young Scientists (Grant No. 42322903). The authors acknowledge the assistance of the Hong Kong Observatory in data acquisition and preprocessing. We sincerely appreciate the reviewers for their valuable suggestions and insightful comments on the original manuscript. We also acknowledge the use of OpenAI's ChatGPT for language refinement in improving the grammar and clarity of an earlier version of this manuscript. All content has been thoroughly reviewed and edited by the authors, who take full responsibility for the
final publication.

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
