# Peer review of "Intra-city Scale Graph Neural Networks Enhance Short-term Air Temperature Forecasting"

_EGUsphere, 2025_

## Author Comment (AC1)

We sincerely thank the reviewers for dedicating their time to reviewing our manuscript and providing constructive comments. These valuable comments have guided us greatly improved the clarity, scientific depth, and overall implications of our study. Our detailed responses are provided below. Referee comments are shown in black, author responses in blue, and newly added manuscript text is shown in *blue italics*.

**Responses to the comments of Referee #1**

We would like to thank the reviewer for her/his valuable comments that help to strengthen the manuscript. Below are detailed responses to her/his comments.

1. The roles of Urban morphology, local circulation, anthropogenic heat release and land use and their impacts on local climate in the Hong Kong should be compared with other regions, that is, can LOI defined in the present work be related to these local background factors? Just for wind or altitude?

**Response**: We sincerely thank the reviewer for the valuable suggestion to strengthen the physical interpretation in our study.

(a) We agree that the LOI essentially reflects the heterogeneous local temperature responses shaped by multiple environmental and meteorological factors. Following this insightful comment, we have expanded the discussion to more explicitly address these factors, including local circulations, land—sea breeze effects, and inland—coastal contrasts, and discussed similar pattern in other cities. We have also selected more representative stations (e.g., inland and interface locations with distinct thermal properties) to better illustrate the associated spatiotemporal dynamics. Accordingly, we have revised Section 3.4 as follows:

Lines 377-413: "Forecast accuracy shows substantial spatial variability, with RMSEs ranging from 0.72 to 1.10 °C (Fig. 6b). Two locations within the most densely developed urban areas show the lowest RMSEs (0.72 and 0.76°C for location 4 and 7, respectively). These locations are surrounded by high-rise buildings in the urban core, where local areas typically have large thermal inertial and reduced ventilation. Location 3 records the highest mean Ta at the airport while demonstrating a relatively low RMSE of 0.82°C. The highest RMSEs are found at locations 14 and 10 where they are in the most inland place (expect the mountain station) and interface of sea water and freshwater reservoir with RMSE of 1.10 and 1.07°C.

[revised manuscript text omitted]

(b) We attempted to use Local Climate Zone (LCZ) data at 100-meter resolution (Demuzere et al., 2022) to associate local factors with the LOI (Fig. R1a). However, no clear pattern was observed (Fig. R1b). This can be partly attributed to the highly localized nature of LCZ classification, which does not adequately represent the influence of surrounding environments that shape local circulations. Moreover, our manual inspection revealed notable classification inaccuracies. For example, the airport site was misclassified as water. These limitations constrain further analysis linking local factors to distinct thermal responses. On the other hand, this also underscores the advantage of the GNN approach, which inherently embeds local characteristics through the data-driven learning process.

**Figure R1.** (a) Local Climate Zone (LCZ) map of Hong Kong. (b) Distribution of daily absolute LOI values at each location, grouped by LCZ type. The middle line within each box represents the median, and the white dot indicates the mean.

(c) We also acknowledge the current limitations of our work and identify potential directions for future research. A more comprehensive spatial investigation of the LOI in relation to its governing local factors and background weather conditions, as well as an evaluation of how the performance gains of graph-based models vary under different settings, would be valuable. We have added the following statement to the revised manuscript: Lines 339-341: "In essence, LOI reflects the heterogeneous local Ta response that are jointly shaped by environmental factors and background weather conditions. A more direct investigation of how these modulate local Ta diurnal variation amplitude, as well as how the performance gains of GSAGE vary across different cities, would be valuable directions for future work."

**Reference:**

Bauer, T. J.: Interaction of Urban Heat Island Effects and Land–Sea Breezes during a New York City Heat Event, https://doi.org/10.1175/JAMC-D-19-0061.1, 2020.

Demuzere, M., Kittner, J., Martilli, A., Mills, G., Moede, C., Stewart, I. D., van Vliet, J., and Bechtel, B.: A global map of local climate zones to support earth system modelling and urban-scale environmental science, Earth System Science Data, 14, 3835–3873, https://doi.org/10.5194/essd-14-3835-2022, 2022.

Oke, T. R., Mills, G., Christen, A., and Voogt, J. A.: Urban Climates, 1st ed., Cambridge University Press, https://doi.org/10.1017/9781139016476, 2017.

Shi, T., Yang, Y., Qi, P., and Lolli, S.: Diurnal variation in an amplified canopy urban heat island during heat wave periods in the megacity of Beijing: roles of mountain–valley breeze and urban morphology, Atmospheric Chemistry and Physics, 24, 12807–12822, https://doi.org/10.5194/acp-24-12807-2024, 2024.

Yang, Y., Guo, M., Wang, L., Zong, L., Liu, D., Zhang, W., Wang, M., Wan, B., and Guo, Y.: Unevenly spatiotemporal distribution of urban excess warming in coastal Shanghai megacity, China: Roles of geophysical environment, ventilation and sea breezes, Building and Environment, 235, 110180, https://doi.org/10.1016/j.buildenv.2023.110180, 2023.

2. About overfitting issues of machine learning models, how did you address or aovid it?

Response: We agree that overfitting is indeed a major concern for machine learning—based models. In this

**Response:** We agree that overfitting is indeed a major concern for machine learning—based models. In this study, we addressed this issue from three perspectives:

(a) **Rigorous model validation:** We used an isolated validation set containing one year of data to tune model graph structures and hyperparameters, preventing overfitting during the hyperparameter tuning phase. We also applied dropout regularization to further mitigate overfitting. The final model evaluation was performed on a test dataset that the model had never accessed during the development process.

- (b) **Mix-n-Scale framework**: Our framework incorporates models with various configurations in an ensemble-based approach. By nature, this ensemble structure allows overfitting to noise in individual models to cancel out, thereby improving generalization performance and reducing parameter uncertainty introduced during training.
- (c) **GSAGE model architecture**: Through comparative evaluation of our GSAGE models and LSTM, we found that local time-series modeling with LSTM tends to overfit the preceding day's pattern. In contrast, GSAGE leverages noise-filtered spatial information as contextual reference, enabling it to better capture more variable  $T_a$  evolution patterns (Fig. 6c).

**3. Station ID can be removed in Figures 2 and 6b for clear show.**

**Response:** Thanks for the suggestion. We agree that the station IDs were not clearly visible in the original figures. For easier reference during the discussion, we decided to keep the IDs but improved their legibility by dynamically adjusting the text color according to the background, making the labels much clearer and easier to read. The figures are revised as follows:

Figure 2. Spatial distribution of (a) mean  $T_a$  and (b) mean diurnal standard deviation (DTSD) over the six-year datasets (basemap © Mapbox). Node colors indicate the magnitude at each location, and numbers denote location IDs using different colors for clearer visualization. The upper and right panels show corresponding values along longitude and latitude, respectively, with the solid line indicating a LOESS-smoothed value.

Figure 7b. Same as Fig. 2 but for Spatial distribution of RMSEs (basemap © Mapbox).

**Responses to the comments of Referee #2**

We would like to thank the reviewer for her/his valuable comments that help to strengthen the manuscript. Below are detailed responses to her/his comments.

**Comments:**

- The authors show that adding GNN-based spatial aggregation improves performance, but it is unclear how
  much of the gain depends on the number, proximity, and connectivity of stations. I suggest including the
  following analyses to provide deeper insight into the value of spatial information for Ta forecasting and to offer
  practical guidance for applying the framework in settings with sparser or differently configured station
  networks.
  - 1.1. Conduct a sensitivity analysis by removing stations one by one and evaluate how forecast performance changes. For example, I would like to know the effectiveness of inclusion of station 2 on station 3 given their distance. This would also help justify the assumption that "clear directed relationships for information propagation from specific 'super-nodes' may not exist" (Lines 236–237).

**Response:** We thank the reviewer for this constructive suggestion with concrete recommendations. We agree that these graph formation analyses would substantially improve our understanding of the impact of spatial information and model configuration. We have conducted the following analyses:

To address the reviewer's first question, we conducted additional ablation experiments to investigate how the number/density and proximity of neighboring stations affect GNN performance.

- (a) Distance-based sensitivity analysis: We systematically varied the number of connected neighbors from 15 down to 1 (i.e., from all neighbors to only the nearest one) to assess how network size/density influences forecast accuracy. Two complementary strategies were tested: (1) nearest-to-farthest, where neighbors are added sequentially outward; and (2) farthest-to-nearest, where neighbors are added inward (e.g., for Station 3, connecting first to the most distant Station 2). The comparison design further allows us to evaluate the importance of proximity in forming connections. The results show that the nearest-to-farthest strategy achieves slightly better overall performance, supporting our assumption that no clear directional influence can be observed from more distant nodes, thereby generally addressing this question.
- (b) Temporal correlation analysis: Furthermore, we agree that Stations 2 and 3, being the most distant nodes, warrant closer examination. Accordingly, we performed a lagged cross-correlation analysis between station pairs to assess whether distant stations exhibit precursory behavior or potentially function as super-nodes influencing others. Specifically, we examined whether stations 2 and 3 exhibit stronger correlations at non-zero time lags, which would suggest a directional information flow. Our analysis shows that stations 2 and 3 have the highest correlation at zero time lag, indicating synchronous behavior rather than a clear lead-lag relationship. Remining pairs also mostly peak their correlation at lag=0. This finding affirms that intra-city scale  $T_a$  dynamics may follow a synchronized pattern, or the timescale is generally below one hourly. We have included these two analyses in the revised manuscript as follows:

Lines 253-266: "Between the two spatial information learning approaches, it is worth noting that GSAGE's simple mean aggregation of neighbor's temporal embeddings outperforms GAT's adaptive attention mechanism, which assigns dynamic weights to neighbors. Although GAT theoretically offers greater flexibility by identifying variable inter-station relationships, and wind vectors are issneluded to provide potential directional cues, this advantage does not manifest here and may instead lead to overfitting issues. This suggests that, at the intra-city scale, there may be no distinct "upstream" information flow or dominant "super-nodes," or, if present, such relationships may occur at shorter timescales. We further examined this hypothesis from a statistical perspective by conducting cross-correlation analyses among station observations with varying time lags to assess whether  $T_a$  changes at one site could precede others. The lag-shifted results show that most interstation correlations peak at zero lag (even for the most distant station pair), with only a few pairs exhibiting slightly higher correlations at  $\pm 1$  h (Fig. S3 in the Supplementary).

Based on the GSAGE model, we further examine how graph construction affects model performance. We find that connecting each station to its nearest neighbor generally yields better performance than linking to the most distant ones (Fig. 4a), even though the latter could capture broader meteorological context and longer-range propagation. This result is consistent with our cross-correlation analysis."

Figure S3. (a) Lag of maximum correlation between station pairs, evaluated over a range of -6 to +6 time steps. In our results, the lags vary between -1 and +1, with most pairs peaking at 0. A lag of 0 indicates that the  $T_a$  time series at the two stations exhibit the highest correlation at the same (synchronized) time step. Positive lags indicate that temperature variations at station i lag behind those at station j, whereas negative lags indicate the opposite. (b) Lagged correlation between stations 2 and 3 (station IDs as shown in Fig. 2), chosen because they are the farthest apart.

**Figure 4a.** Variation of RMSE with the number of neighboring nodes used to form edge connection under two strategies. The solid curve denotes the mean RMSE, and the shaded area represents the standard deviation across models trained with different hyperparameter settings."

1.2. Verify whether the observed performance gains arise from the GNN's relational structure rather than simply having access to neighbor data. Consider comparing with a simpler baseline where neighbor embeddings are concatenated or averaged and fed directly to the final prediction layer without using GNN.
Response: We greatly appreciate the reviewer's insightful observation regarding the model structure. We would

like to clarify that with a single layer of GSAGE spatial aggregation, the operation is indeed very similar to what the reviewer suggested, essentially concatenating the node's own embedding with the averaged embedding of its

neighbors. We therefore have added an explicit description of the GSAGE model in Section 2.2 (Problem Formulation and DL Models).

Furthermore, we examine whether a two-layer GNN architecture, with greater capacity to capture higherorder information, could achieve improved performance, which could be attributed to the inherent structural advantages of GNNs. Our experiments demonstrate that the model's performance improves substantially, with RMSE decreasing from approximately 1.01 to 0.96 when increasing the network depth to two layers. We revise the manuscript as follows:

Lines 131-133: "GSAGE adopts an undirected graph structure with uniform neighbor weighting via mean aggregation, which was selected over max/min pooling in preliminary testing."

Lines 267-273: "Another key structural factor is graph depth, which determines how many hops of neighbor information each node can access. We observe more than a 5% RMSE reduction when using two GNN layers compared to a single layer (Fig. S4 in the Supplement). Although a single layer with full connectivity can theoretically access the entire graph, adding a second layer does not expand the receptive field but introduces additional nonlinearity and feature-transformation capacity, potentially improving model expressiveness. However, deeper architectures that repeatedly aggregate neighbor information do not provide further gains in our case and may instead lead to over-smoothing, making node representations less distinguishable."

**Figure S4.** Variation of RMSE with the number of neighboring nodes used to form edge connections, classified by graph depth (one-layer and two-layer GNNs). The solid curve denotes the mean RMSE, and the shaded area represents the standard deviation across models trained with different hyperparameter settings.

2. Discuss why including global predictors appears to worsen GSAGE performance (Fig. S1) and explain the rationale for keeping them.

**Response:** We thank the reviewer for this insightful comment, which helps improve the clarity of our study.

(a) Global predictors appear to worsen model performance, likely because their single-point values lack spatial patterns and gradients (e.g., advection or trough/ridge structures), thereby introducing minimal useful information and potentially adding noise. Indeed, in an in-preparation study where these variables are incorporated as spatial fields, we find substantial improvement (Wang and Yang, 2025). We have addressed this point in Section 3.2 as follows:

In lines 279-282: "This is likely because such variables are more physically meaningful when their spatial patterns and gradients are represented (e.g., pressure gradients that drive large-scale flows or synoptic features such as troughs and ridges). When incorporated as single-point values, they provide limited information and may even introduce noise."

(b) The final performance reported in this study is based on the Hyper-GSAGE model trained without global predictors. These predictors are included only for exploratory purposes in the ablation experiments to assess their potential contributions. For clarity, we have revised the corresponding description at Section 2.1 (Datasets) as follows:

In lines: "It is worth noting that the final model performance is reported based on training without global predictors, as their inclusion did not yield improvement. These variables are retained only for ablation analysis (Section 3.2; Fig. 4b) to illustrate their potential influence on model performance."

**Response:** Thanks for this very constructive suggestion.

(a) We agree that although Figure S1 demonstrates the impact of ensemble size, additional clarity is needed regarding hyper-GSAGE's sensitivity to search space size, hyperparameter ranges, and computational scaling. We have now included these experiments with the following results:

Lines 236-240: "Its performance remains highly stable across different ensemble sizes, achieving optimal accuracy when incorporating the top ~10% of models from validated pool (10 out of 100; Fig. S1 in the Supplement). Building such a pool typically requires around 50 hyperparameter trials drawn from a broad

initial search space (Fig. S2 in the Supplement), which can be completed within 10 hours on a single RTX 4090 GPU. Once finalized, the model generates forecasts within seconds, enabling efficient real-time applications.

**Figure S1.** Variation in Hyper-GSAGE performance with different ensemble member sizes. Each column represents the performance of an individual GSAGE model, sorted by ascending validation error. The green line denotes the Hyper-GSAGE performance with a corresponding number of best models. It should be noted that the observed increase in RMSE with large ensemble sizes beyond 13 is primarily due to the inclusion of failure models. Conducting additional trials within the optimal hyperparameters range generally achieves a better performance, and this graph is only for illustrative purposes.

**Figure S2.** Variation in hyper-GSAGE performance (ensemble size of 10 members) across different hyperparameter search runs. The blue line represents searches initiated from a very broad search space (as defined in Table S1), while the orange line represents searches within a refined space based on initial search results. Performance stabilizes at fewer than 50 runs, with less than 1% variation beyond this point.

(b) Regarding hyperparameter sensitivity, we have included an analysis of the top 5 model configurations from our study, along with a discussion of the underlying rationale for key hyperparameters, to serve as a reference for similar applications. For the graph formation details and their topologies, we have addressed these in our response to Comment 1, which is relevant to Fig. 4a and Fig. S4. We have added these hyper-parameter analyses as follows:

In the Supplement: "Table S1 lists the top five model configurations (out of 100 trials) ranked by their validation performance. Although the search range for time lag was set up to 200, the optimal configurations tend to select

relatively short lags. This suggests that while incorporating temporal sequences benefits the model, excessively long input windows (though theoretically containing more information) may introduce redundant or noisy signals that ultimately degrade performance. Similar observations have been reported in a purely time-series forecasting study (Wang et al., 2024).

The optimal number of GNN layers is generally two, indicating that moderate spatial aggregation effectively captures global spatial dependencies, whereas deeper GNNs may lead to over-smoothing across locations. Regarding the number of neighbors, the results show that models typically perform better when incorporating a larger number of spatial connections, implying that richer inter-station relationships enhance representational learning."

Table S1. Model configurations with top five validation performances (the brackets [] indicate the search range for each parameter).

| Validation | Time lag | Hidden    | GNN layer | Neighbour | learning              | parameter |
|------------|----------|-----------|-----------|-----------|-----------------------|-----------|
| RMSE       | [1, 200] | dimension | [1, 3]    | size      | rate                  | number    |
|            |          | [10, 200] |           | [1, 15]   | [5e-5, 1e-3]          |           |
| 0.903      | 44       | 116       | 2         | 15        | 2.17×10 -4 | 164959    |
| 0.915      | 30       | 135       | 2         | 12        | 1.61×10 -4 | 222757    |
| 0.916      | 32       | 109       | 2         | 12        | 1.61×10 -4 | 145849    |
| 0.918      | 5        | 171       | 2         | 13        | 1.13×10 -4 | 356029    |
| 0.922      | 2        | 62        | 3         | 12        | 1.93×10 -4 | 55869     |

5. The explanation of the spatiotemporal pattern of forecast performance in Section 3.4 could be clearer. While the authors report RMSEs, local Ta variability, and site characteristics, they do not clearly connect these characteristics to the observed performance differences. For instance, the statement that "the pronounced diurnal contrast can be primarily attributed to solar radiation-induced perturbations and consequent atmosphere-land interactions, highlighting the inherent challenges in capturing daily peak values" (Lines 323 – 325) notes the difficulty but does not explain why these processes make the peak harder to forecast. I recommend revising this section to make those linkages more explicit.

Response: We thank the reviewer for this constructive suggestion. We agree that the original discussion lacked explicit mechanistic explanations for the observed performance differences. To address this, we have conducted additional quantitative analyses, including time-series persistence analysis, to better characterize the underlying  $T_a$ evolution dynamics from both diurnal and seasonal perspectives. The revised Section 3.4 now provides clearer linkages between underlying  $T_a$  dynamics and forecast difficulty. The key revisions are as follows: Lines 355-375: "Following the successful development of the Hyper-GSAGE model, we further evaluate its spatiotemporal forecast performance to elucidate the variability and underlying dynamics of prediction errors. The results reveal a pronounced diurnal contrast, with RMSEs increasing during the daytime and peaking between 10:00-14:00 (1.27-1.40 °C), coinciding with the warmest period of the day (Fig. S6 in the Supplement). In contrast, nighttime forecasts, particularly between 00:00–04:00, exhibit the lowest RMSEs (0.61–0.63 °C). This pattern remains consistent even when RMSEs are normalized by the mean hourly  $T_a$  of the corresponding periods (Fig. S7 in the Supplement). The distinct diurnal variation in forecast skill can be primarily attributed to differences in  $T_a$ evolution dynamics between day and night. During daytime, solar radiation-induced surface heating and subsequent atmosphere-land interactions introduce strong perturbations, amplifying  $T_a$  variability and increasing forecast difficulty. After sunset, however,  $T_a$  evolves more smoothly under stable boundary-layer conditions, resulting in reduced variability and lower forecast errors. This diurnal contrast is further supported by the autocorrelation analysis (Fig. S8a in the Supplement), which indicates substantially higher nighttime persistence

(~0.94) compared with daytime, particularly around 12:00–14:00 when persistence reaches a minimum (~0.84) at the 1-hour lag. A similar contrast is also observed for the 1-day lag (same hour on the previous day), with persistence values of ~0.75 at night and ~0.57 during midday. Collectively, these results demonstrate that daytime  $T_a$  variability is more dynamic and thus inherently less predictable from a statistical perspective. Seasonally, both summer and winter exhibit elevated forecast errors (RMSEs of 1.00 °C and 0.92 °C, respectively). While summer remains relatively stable under the control of the subtropical high-pressure system (24-h lag autocorrelation of 0.37 in summer, compared with -0.01 in winter; Fig. S8b in the Supplement), stronger radiative forcing and turbulent energy exchange within a more energetic atmosphere likely contribute to greater short-term  $T_a$  variability at hourly scales. This is reflected by the lower 1-hour autocorrelation (0.48 in summer compared with 0.72 in winter; Fig. S8b in the Supplement), presenting greater challenges for short-range forecasting.

Figure S8. (a) Diurnal and (b) seasonal variations in the autocorrelation coefficients of observed  $T_a$  at 1-hour and 24-hour time lags (dashed line), together with the corresponding RMSE of Hyper-GSAGE forecasts (solid line). Shaded areas denote one standard deviation across all stations. Both lag correlations exhibit a pronounced midday minimum, indicating diminished  $T_a$  persistence during the daytime, and enhanced persistence at nighttime. RMSE varies inversely with autocorrelation coefficients, indicating greater forecast uncertainty during periods of lower persistence."

**Minor comments:**

1. Latitude and longitude swapped in Table 1.

**Response:** The table has been revised in the revised manuscript.

Since GAT and GSAGE themselves are GNNs without LSTM encoding, the labels or figure captions could be clarified to avoid misunderstanding. For example, Fig. 1 may read as though you are comparing a standalone GNN with an LSTM, rather than GNNs applied on top of LSTM embeddings.

**Response:** We would like to thank the reviewer for this helpful suggestion to improve the clarity of the manuscript. We have revised Figure 1 and updated its caption to better illustrate the model structure, as follows:

Figure 1. Schematic of the modelling framework. (a) Spatial distribution of weather observation stations across Hong Kong (basemap © Mapbox), with location IDs labeled. The edges between stations represent the schematic GNN structure, showing nine connections per node. (b, c) Conceptual diagram comparing the local time-series modeling approach with the graph-based approach, in which LSTM-based temporal embeddings are spatially aggregated using GNN across neighboring stations. (d) Overview of the Mix-n-Scale framework, which leverages intra-city observations using diversely configured GNNs."

- 3. Line 211: revise "the mean Ta patterns is ..." to "the mean Ta patterns are ...". **Response:** The statement has been revised in the revised manuscript.
- 4. Line 266: revise "significantly outperform than ..." to "significantly outperform the ...". **Response:** The statement has been revised in the revised manuscript.